# Using deep learning to identify inherited retinal diseases based on wide-field retinal imaging data

Leo Joskowicz[1], Tim Buchbinder[1], Eldan Chodorov[1], Assaf Hoogi [2], Katherine Matos[3], Antonio Rivera[3], Dror Sharon[3], Eyal Banin[3], Jaime Levy [3]*

**1** School of Computer Science and Engineering, The Hebrew University of Jerusalem, Jerusalem, Israel, **2** School of Computer Science, Ariel University, Ariel, Israel, **3** Department of Ophthalmology, Hadassah Medical Center, Faculty of Medicine, The Hebrew University of Jerusalem, Jerusalem, Israel

* levjaime@gmail.com

## Abstract

### Objective

To evaluate a novel image-based deep learning method for the automated identification of inherited retinal diseases (IRDs) and to explore the feasibility of predicting selected causative gene groups using a multimodal analysis of wide-field fundus autofluorescence (FAF) and pseudocolor fundus (pCF) images.

### Design

The method was evaluated using a retrospective dataset of patient studies containing FAF and pCF images, as well as genetic tests for IRD.

### Participants

Patients with confirmed IRD for which both wide-field FAF and pCF images and genetic tests for IRD performed at Hadassah University Medical Center were included. The dataset consisted of 409 patients (330 patients with IRD with the 25 most commonly affected genes in our population and patients without IRD, and 79 patients without IRD).

### Methods

Nine EfficientNet-V2-m convolutional neural networks were trained for the following three classification tasks: a binary IRD vs. non-IRD classification, and classification into two groups of five causative genes (Groups 1 and 2). For each task, three models were trained on the FAF images only, the pCF images only, and both the FAF and pCF images. The performance of the models was then evaluated and compared using 5-fold cross-validation.

**Data availability statement:** All data supporting the findings of this study are available within the manuscript and in supplementary materials. Due to patient privacy and ethical restrictions, raw retinal images cannot be made publicly available. However, all aggregated performance metrics, confusion matrices, model architecture details, and hyperparameters are provided within the manuscript and supplementary materials. The code is available at: https://github.com/timshik/Deep-learning-based-identification-of-inherited-retinal-disease-using-wide-field-retina-imaging.

**Funding:** The author(s) received no specific funding for this work.

**Competing interests:** The authors have declared that no competing interests exist.

## Main outcome measures

Accuracy, precision, F1 scores, AUC, and confusion matrices.

## Results

The multimodal classification models that were trained on both the FAF and pCF images yielded the best results. The binary classification model had a mean (±SD) accuracy of $0.95 \pm 0.01$, a mean precision of $0.92 \pm 0.01$, and a mean F1 score of $0.90 \pm 0.02$. The Group 1 classification model had a mean accuracy of $0.92 \pm 0.03$, a mean precision of $0.93 \pm 0.03$, and a mean F1 score of $0.89 \pm 0.03$. Finally, the Group 2 classification model had a mean accuracy of $0.85 \pm 0.03$, a mean precision of $0.87 \pm 0.04$, and a mean F1 score of $0.83 \pm 0.04$.

## Conclusions

Our results indicate that determining whether a patient has IRD can be performed with high accuracy within this retrospective cohort based on FAF and pCF images using image-based deep learning classifiers. This image-based approach may assist clinicians during the patient's initial visit by providing decision support prior to genetic testing. It may also help prioritize patients for genetic workup, particularly in settings in which genetic testing is not readily available. Further prospective and external validation is required before clinical implementation.

## 1 Introduction

Inherited retinal dystrophies (IRDs) represent a heterogeneous and clinically complex group of genetic disorders characterized by progressive degeneration of retinal photoreceptors, leading to variable degrees of visual impairment and, in many cases, eventual blindness. These diseases encompass a broad phenotypic spectrum including symptoms such as night blindness, peripheral vision loss, color vision defects, and central visual acuity decline, reflecting the diverse retinal cell types and pathways affected [1]. The genetic underpinnings of IRDs are exceptionally diverse, with pathogenic variants identified in over 270 genes to date, spanning multiple inheritance patterns and molecular mechanisms [2–5]. This genetic and phenotypic heterogeneity complicates diagnosis and patient management, as mutations in a single gene can produce distinct clinical presentations, and conversely, similar phenotypes may arise from mutations in different genes [3,6].

Epidemiologically, IRDs affect approximately 1 in 1,000–3,000 individuals worldwide, amounting to millions of affected patients [7–9]. Additionally, carrier rates for autosomal recessive IRD variants are remarkably high, underscoring the substantial genetic burden within the global population [9]. Given the progressive nature of these diseases and their profound impact on quality of life, early and accurate diagnosis is critical. Although genetic testing remains the definitive diagnostic tool and enables precise molecular classification, it is not universally accessible due to financial,

infrastructural, and logistical constraints [10,11]. Consequently, clinical diagnosis frequently relies on detailed ophthalmic examinations and multimodal retinal imaging, which, while valuable, often lack specificity and are subject to interobserver variability [7,12,13]. Delayed or inaccurate diagnosis can hinder timely intervention, genetic counseling, and eligibility for emerging gene therapies and clinical trials [14,15].

Recent advances in retinal imaging, including wide-field color fundus photography and fundus autofluorescence (FAF), have substantially enhanced the visualization of retinal pathology in IRDs, enabling improved phenotypic characterization and monitoring [16]. However, interpretation of these images demands specialized expertise and remains resource-intensive, creating barriers in underserved regions and contributing to a diagnostic bottleneck [17].

In parallel, artificial intelligence (AI), particularly deep learning approaches, have revolutionized image-based disease detection in ophthalmology, demonstrating high accuracy in screening for conditions such as diabetic retinopathy, age-related macular degeneration, and glaucoma [18–21]. Preliminary studies applying deep learning to IRDs have shown promise in distinguishing specific subtypes from retinal images, mostly focusing on FAF or optical coherence tomography (OCT) modalities [22–25]. Yet, these efforts have been limited by small datasets, narrow classification tasks, and an absence of robust models for binary IRD identification or direct prediction of underlying genetic causes from imaging alone.

Despite the increasing availability of genetic testing, access remains limited in many regions due to cost, infrastructure, or lack of awareness. An accurate, image-based tool that can assist in the early identification of IRDs, prior to or when genetic testing is not readily available, could substantially reduce diagnostic delays and support timely clinical decision-making. This study addresses an important clinical challenge by evaluating a novel deep learning–based method for classifying IRD versus non-IRD using multimodal wide-field retinal images and investigating the potential to predict causative gene groups directly from imaging data. Such an approach may be helpful in resource-limited settings and could help prioritize patients for confirmatory genetic workup.

This study aims to develop and rigorously evaluate a novel deep learning–based approach leveraging multimodal Optos wide-field FAF and pseudocolor fundus (pCF) images for automated IRD detection. Additionally, we explore the feasibility of predicting IRD-related genetic groups directly from imaging data and assess whether combining imaging modalities enhances classification performance compared to single-modality analysis.

## 2 Methods

### 2.1 Dataset

This retrospective cohort study was approved by the Hadassah Medical Center Ethics Committee (HMO 382−19) and conducted in accordance with the Declaration of Helsinki. Clinical records from patients with genetically confirmed inherited retinal dystrophies (IRDs) who attended the Retina Clinic at Hadassah University Medical Center (Jerusalem, Israel) between January 2019 and April 2023 were collected and anonymized [26]. The local ethics committee waived the requirement for informed consent due to the retrospective and anonymized nature of the data. Data were accessed starting January 23, 2023, and analyzed through May 25, 2024. All data supporting the findings of this study are available within the manuscript and in supplementary materials. Due to patient privacy and ethical restrictions, raw retinal images cannot be made publicly available. However, all aggregated performance metrics, confusion matrices, model architecture details, and hyperparameters are provided within the manuscript and supplementary materials. The code is available at: https://github.com/timshik/Deep-learning-based-identification-of-inherited-retinal-disease-using-wide-field-retina-imaging.

Each patient visit included examinations of one or both eyes, with acquisition of wide-field fundus autofluorescence (FAF) and pCF images, alongside genetic testing results. Images were captured using Optos California, Optos 200T×, and Optos Silverstone systems (Optos PLC, UK). Genetic results were available for all IRD cases with imaging. For training the model, we selected the 25 most commonly affected genes in our cohort to maximize data representation (Table 1).

**Table 1. Distribution of the 330 patients with IRD (full information in S1Table).**

| Short name | Causative gene | Number of patients | Number of patient exams | Patient age | | |
|---|---|---|---|---|---|---|
| | | | | Mean | Min | Max |
| A | ABCA4 | 57 | 158 | 39 | 12 | 82 |
| B | BEST1 | 9 | 27 | 21 | 12 | 39 |
| C | CACNA1F | 6 | 12 | 34 | 8 | 63 |
| D | CERKL | 6 | 12 | 43 | 20 | 67 |
| E | CHM | 12 | 28 | 37 | 19 | 72 |
| F | CNGA3 | 13 | 33 | 32 | 6 | 62 |
| G | CNGB3 | 7 | 14 | 36 | 21 | 46 |
| H | CRB1 | 12 | 29 | 30 | 6 | 51 |
| I | DHDDS | 16 | 40 | 45 | 29 | 81 |
| J | EYS | 18 | 49 | 55 | 28 | 74 |
| K | FAM161A | 28 | 79 | 49 | 9 | 87 |
| L | GUCY2D | 6 | 14 | 48 | 38 | 65 |
| M | KIZ | 9 | 26 | 40 | 19 | 84 |
| N | MAK | 10 | 23 | 62 | 31 | 84 |
| O | MY07A | 7 | 15 | 32 | 17 | 47 |
| P | NR2E3 | 12 | 35 | 35 | 13 | 74 |
| Q | OPN1LW-OPN1MW | 9 | 28 | 36 | 12 | 67 |
| R | PRPF31 | 8 | 26 | 38 | 19 | 64 |
| S | PRPH2 | 8 | 24 | 60 | 48 | 74 |
| T | RDH12 | 11 | 35 | 30 | 6 | 53 |
| U | RHO | 11 | 35 | 54 | 3 | 94 |
| V | RPE65 | 5 | 16 | 28 | 6 | 57 |
| W | RPGR | 22 | 48 | 35 | 14 | 69 |
| X | TRPM1 | 8 | 16 | 18 | 12 | 31 |
| Y | USH2A | 20 | 54 | 52 | 31 | 71 |
| Total IRD | Genes | 330 | 876 | 40 | 3 | 94 |
| Group 1 (A,P,T,X,Y) | ABCA4, NR2E3, RDH12, TRPM1, USH2A | 108 | 298 | 39 | 6 | 82 |
| Group 2 (B,E,F,H,K) | BEST1, CHM, CNGA3, CRB1, FAM161A | 74 | 196 | 38 | 6 | 87 |

These 25 genes accounted for approximately 63% of all genetically confirmed IRD cases evaluated at our center during the study period, highlighting their substantial local relevance and their suitability for inclusion in the model.

After quality control excluding images with poor resolution, corruption, or blurring, the final dataset comprised 330 IRD patients with 876 imaging exams. Patient demographics and gene distribution are summarized in Table 1. Additionally, 79 patients without IRD were included as controls, comprising 24 patients with other ocular diagnoses and 55 patients without retinal pathology, totaling 152 control images (Table 2).

## 2.2 Image preprocessing

To ensure consistency, all FAF and pCF images were preprocessed to standardize size, scale, and field of view. Noise such as text annotations and partial eye occlusions (eyelids, eyelashes) at the borders were removed (see Fig 1). Image intensities were normalized, and local regions were cropped centrally with varying factors during training and validation to augment data variability (details in Supplemental Materials).

 

**Table 2. Distribution of the 79 patients in the non-IRD group, including 24 patients with a diagnosis other than IRD and 55 patients with a non-retinal ocular disease.**

| Diagnosis | Number of patients |
|---|---|
| Retinal tear/hole after laser treatment | 2 |
| Pars planitis | 3 |
| Choroidal nevus | 4 |
| Epiretinal membrane | 3 |
| Hypertensive retinopathy | 1 |
| Diabetic retinopathy | 1 |
| Age-related macular degeneration | 1 |
| Myopic choroidal neovascularization | 1 |
| Retinal detachment | 1 |
| Lattice degeneration of retina | 2 |
| Optic atrophy | 2 |
| White dot syndrome | 1 |
| Central serous chorioretinopathy | 2 |
| **Total** | **24** |
| No retinal pathology | 55 |
| **Total non-IRD** | **79** |

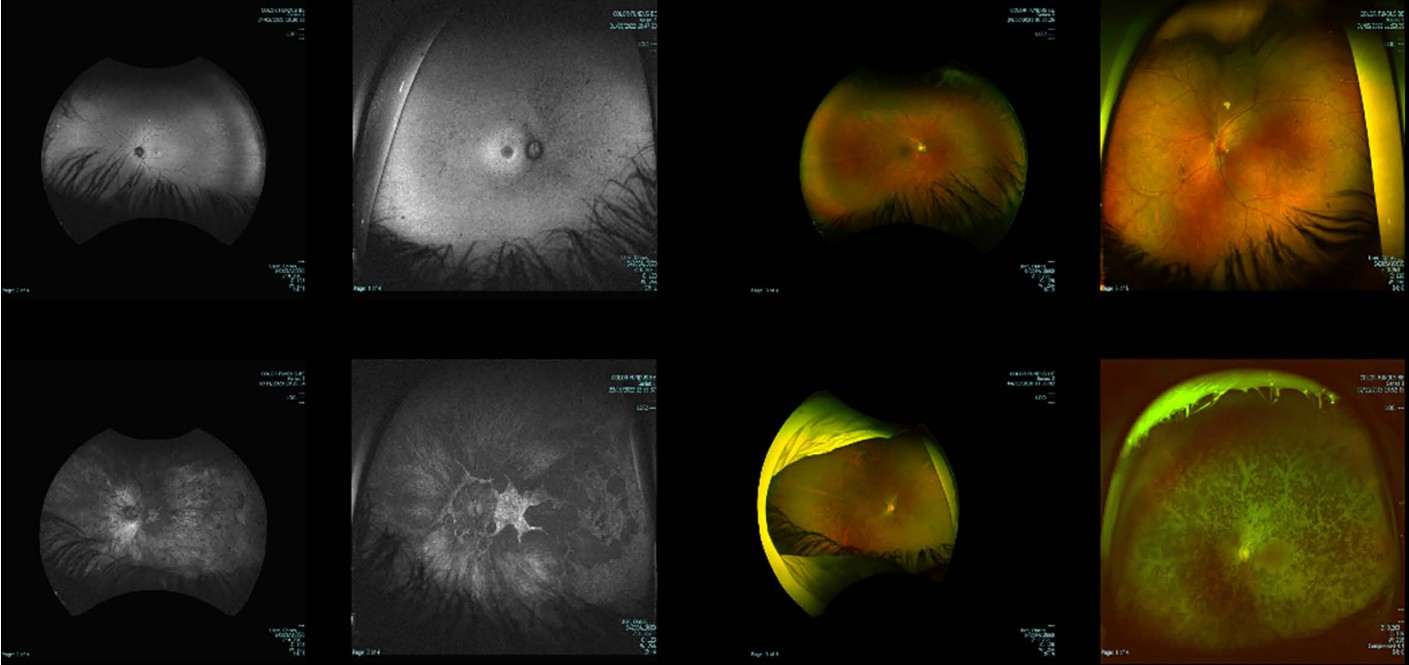

**Fig 1. Representative fundus autofluorescence (FAF, left) and color fundus (pCF, right) images illustrating variability in the dataset, including differences in intensity, magnification, text annotations, and partial occlusion by eyelashes or eyelids.** These images are intended to show the diversity of imaging conditions rather than the difficulty of classification or the underlying diagnosis.

## 2.3 Classification tasks and gene grouping

We defined three binary classification tasks:

1. IRD vs. non-IRD: Identifying whether a patient has IRD regardless of gene mutation.

2. Group 1 gene classification: Classifying whether an IRD patient harbors a mutation in any of five specific genes (ABCA4, NR2E3, RDH12, TRPM1, USH2A).

3. Group 2 gene classification: Classifying whether an IRD patient harbors a mutation in any of another five genes (BEST1, CHM, CNGA3, CRB1, FAM161A).

Grouping genes was necessary because some individual genes had insufficient patient numbers for robust classification (e.g., only 5 patients for RPE65). Groups were defined empirically to balance sample size across classes and optimize model performance in preliminary experiments, rather than based on biological or clinical similarity. This approach allowed us to explore gene-level prediction in a proof-of-concept framework while avoiding overfitting due to sparse data.

## 2.4 Deep learning models

For each classification task, we trained three separate models using different input modalities: FAF images alone, pCF images alone, and a combination of both FAF and pCF images. To prevent data leakage arising from multiple images belonging to the same patient, all dataset partitions were performed at the patient level. Thus, all images from a given patient were assigned exclusively to the training, validation, or testing set. This ensured that the CNNs were evaluated on entirely unseen patients. We used the EfficientNet-V2-m convolutional neural network architecture, selected for its strong performance with relatively small and imbalanced datasets [27]. The final fully connected layer was replaced with two layers, followed by batch normalization and ReLU activation (see Fig. 2).

In addition, to mitigate the impact of limited sample size, we applied standard data-augmentation techniques as random vertical and horizontal flips, as well as translations up to 15 pixels in both directions. These augmentations were applied probabilistically (30% chance each). Random center cropping between 47.5% and 72.5% of the image size was also performed during training.

Models were initialized with ImageNet-pretrained weights and fine-tuned using the Adam optimizer with a learning rate of 0.001 for up to 100 epochs [28,29]. Early stopping and batch normalization were employed to prevent overfitting. A

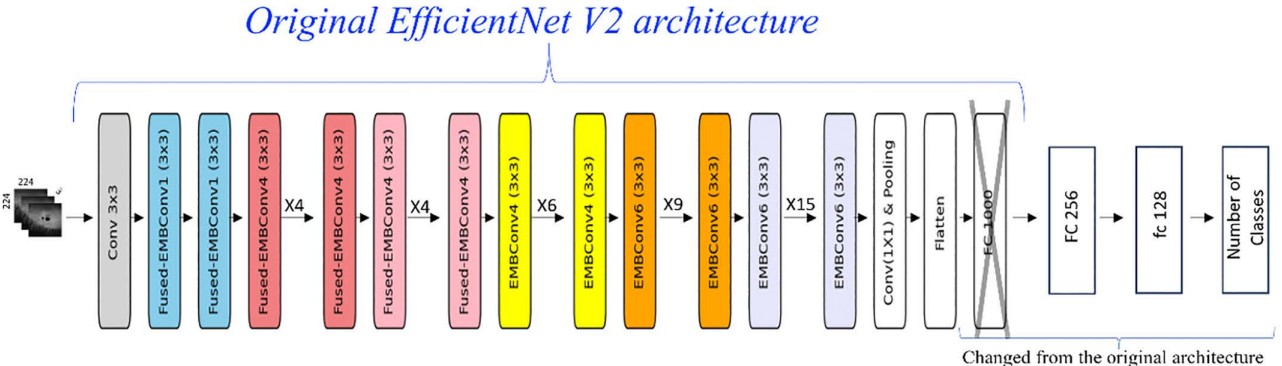

**Fig 2. The modified EfficientNet-V2 architecture used in this study, in which the last fully connected layer (FC 1280) was replaced by two fully connected layers (FC 256 and FC 128).** The number of output neurons in the last fully connected layer was determined by the number of classes. Layers FC 256 and FC 128 are the combination of a fully connected layer, batch normalization, and ReLU activation [28].

weighted cross-entropy loss function balanced performance across class imbalances. Hyperparameters, including learning rate, augmentation probabilities, cropping range, and regularization parameters, were selected based on five-fold cross-validation, and final values were chosen to optimize validation performance and generalization. Each fold maintained strict patient-level separation across training and testing subsets. All images were resized to 224×224 pixels for model input. PCF images retained their RGB format, FAF grayscale images were duplicated across RGB channels, and for combined models, a grayscale-converted pCF image was input into one channel while FAF images filled the other two channels. All information is presented in S2Table.

## 2.5 Experimental setup

We conducted three main experiments, each using five-fold cross-validation:

- Experiment 1: Multi-class classification of IRD versus non-IRD across all patients. This was a binary classification task including all 330 IRD patients and 79 non-IRD patients (mean test set size ~206 studies per fold).

- Experiment 2: Five-class classification task including only IRD patients belonging to Group 1 (ABCA4, NR2E3, RDH12, TRPM1, USH2A). A total of 108 patients (298 images) were included. Each patient was assigned to one of the five gene classes. (mean test set size ~60 studies per fold).

- Experiment 3: Five-class classification task including only IRD patients belonging to Group 2 (BEST1, CHM, CNGA3, CRB1, FAM161A). A total of 74 patients (196 images) were included. Each patient was assigned to one of the five gene classes. (mean test set size ~39 studies per fold).

## 2.6 Performance evaluation and statistical analysis

Model performance was assessed using accuracy, precision, recall, F1 score, area under the receiver operating characteristic curve (AUC), and normalized confusion matrices. Metrics were computed per fold and averaged over five folds. Statistical significance of performance differences was evaluated using paired Wilcoxon tests, with p-values <0.05 considered significant [30].

The implemented method is available at link: https://github.com/timshik/Deep-learning-based-identification-of-inherited-retinal-disease-using-wide-field-retina-imaging.

## 2.7 Data partitioning strategy and held-out testing

All experiments were performed using strict patient-level five-fold cross-validation. In each fold, approximately 80% of patients were used for training and 20% were held out for testing. Each fold functions as a strictly held-out test set. Crucially, the splitting was performed at the patient level (rather than the image level) to ensure zero data leakage between patients; images from a specific patient never appear in both the training and the held-out test fold simultaneously.

Within the training set of each fold, a random 10% subset of patients was used as a validation set for early stopping and hyperparameter tuning. This validation set was also separated at the patient level from the remaining training subset.

For Experiments 2 and 3, the test set included only IRD patients belonging to the relevant gene group under evaluation. Each of these experiments constituted a multi-class classification task with five classes corresponding to the five genes in the group. No patients from other genes were included in the test set for these experiments.

This design ensures that model performance reflects true generalization to unseen patients rather than image-level memorization.

No external validation cohorts was available, and all results reflect internal cross-validation performance within a single-center dataset.

# 3 Results

Table 3 summarizes the results of the three experimental studies, and Figs 3-5 show the normalized confusion matrices for each study for the three imaging modalities. In our analysis, we adopted a hybrid evaluation strategy that integrates metrics commonly reported in artificial intelligence and machine learning research (e.g., accuracy, precision/ positive predictive value, F1 score, area under the curve (AUC)) with elements that align with the expectations of clinical and medical journals, such as class-wise accuracy [31]. Calibration measures and decision curve analysis were not performed at this stage, as the current analysis did not extend to clinical deployment scenarios. Similarly, comparison with human experts was outside the scope of this phase but remains an important consideration for future validation.

**Experimental Study 1**: our binary classification of IRD vs. non-IRD based on the combination of FAF and pCF images yielded a mean accuracy of $0.95 \pm 0.01$. We found no significant difference in performance between the combined FAF and pCF model and the model based on FAF alone; however, both of these models had significantly higher accuracy than the model based on pCF alone.

**Experimental Study 2**: our classification of Group 1 genes based on the combination of FAF and pCF images yielded a mean accuracy of $0.92 \pm 0.03$, a mean precision of $0.93 \pm 0.03$, a mean F1 score of $0.90 \pm 0.03$, and a mean AUC of $0.97 \pm 0.02$. We found no significant difference in accuracy between the combined FAF and pCF model and the model based on FAF alone in terms of accuracy, likely due to the high prevalence of gene A, in which the FAF-only model had slightly better performance than the combined model. However, a statistically significant improvement in the F1 metric was observed for the combined model compared to the FAF-only model. Furthermore, both the combined FAF and pCF model and the FAF-only model performed significantly better than the pCF-only model with respect to accuracy, precision, and F1 scores.

**Table 3. Results of the three studies for the three imaging modalities. Values in bold indicate the best result; while underlined values indicate statistical significance in Studies 2 and 3 (not applicable for Study 1). Data are presented as the mean ± SD values from the five-fold cross-validation.**

| Study classification | Imaging modality | Accuracy | Precision | F1 | AUC | Individual class accuracy | | | | |
|---|---|---|---|---|---|---|---|---|---|---|
| **Study 1 IRD/non-IRD** *n*=206 | | | | | | **IRD** | | | **Non-IRD** | |
| | **FAF+pCF** | **0.95 ±0.01** | 0.92 ±0.08 | **0.90 ±0.02** | 0.96 ±0.03 | **0.96 ±0.02** | | | 0.91 ±0.08 | |
| | **FAF** | **0.95 ±0.01** | **0.93 ±0.04** | **0.90 ±0.02** | **0.98 ±0.01** | 0.95 ±0.01 | | | **0.93 ±0.04** | |
| | **pCF** | 0.93 ±0.02 | 0.92 ±0.01 | 0.88 ±0.02 | **0.98 ±0.02** | 0.94 ±0.02 | | | 0.92 ±0.04 | |
| **Study 2 Group 1** *n*=60 | | | | | | **A** | **P** | **T** | **X** | **Y** |
| | **FAF+pCF** | **0.92 ±0.03** | **0.93 ±0.03** | <u>0.90 ±0.03</u> | **0.97 ±0.02** | 0.94 ±0.05 | 0.89 ±0.11 | **0.90 ±0.1** | 1 ±0.0 | **0.88 ±0.08** |
| | **FAF** | 0.91 ±0.03 | 0.92 ±0.04 | 0.84 ±0.08 | **0.97 ±0.02** | **0.96 ±0.04** | **0.93 ±0.07** | 0.80 ±0.29 | 1 ±0.0 | 0.82 ±0.1 |
| | **pCF** | 0.78 ±0.07 | 0.81 ±0.04 | 0.64 ±0.13 | 0.93 ±0.01 | 0.92 ±0.05 | 0.66 ±0.26 | 0.41 ±0.25 | 0.54 ±0.31 | 0.77 ±0.21 |
| **Study 3 Group 2** *n*=39 | | | | | | **B** | **E** | **F** | **H** | **K** |
| | **FAF+pCF** | <u>0.85 ±0.03</u> | <u>0.87 ±0.04</u> | <u>0.83 ±0.04</u> | <u>0.94 ±0.03</u> | **0.89 ±0.1** | **0.83 ±0.18** | **0.85 ±0.18** | **0.72 ±0.19** | 0.89 ±0.07 |
| | **FAF** | 0.78 ±0.07 | 0.82 ±0.09 | 0.75 ±0.1 | 0.93 ±0.04 | 0.76 ±0.28 | 0.76 ±0.22 | 0.84 ±0.21 | 0.60 ±0.26 | 0.86 ±0.07 |
| | **pCF** | 0.74 ±0.11 | 0.82 ±0.07 | 0.68 ±0.12 | 0.89 ±0.08 | 0.61 ±0.33 | 0.74 ±0.22 | 0.57 ±0.32 | 0.61 ±0.33 | **0.94 ±0.07** |

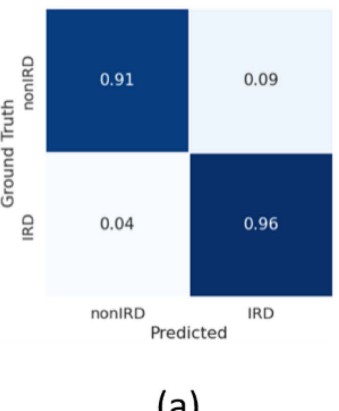 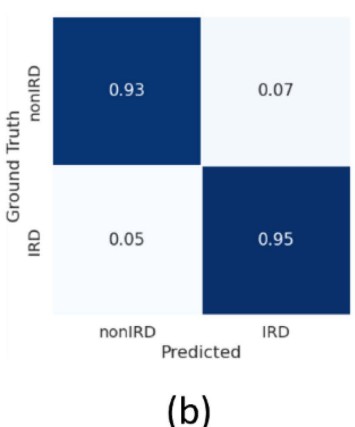 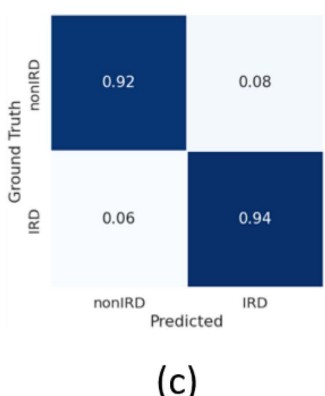

(a) (b) (c)

**Fig 3. Normalized confusion matrices for Study 1 (IRD vs. non-IRD classifiers) based on the combined FAF and pseudocolor model (a), the FAF-only model (b), and the pseudocolor-only model (c).** For each model, the columns indicate the ground truth, and the rows indicate the model's prediction.

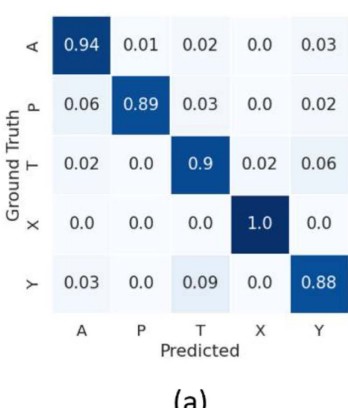 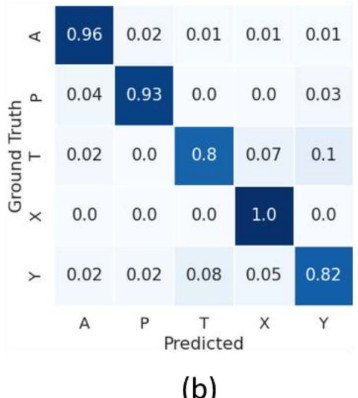 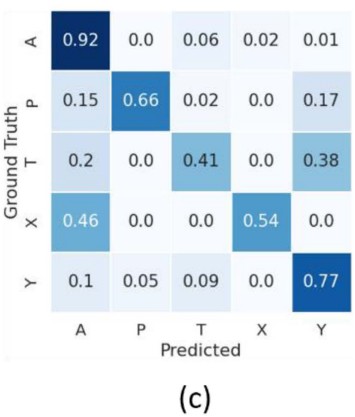

(a) (b) (c)

**Fig 4. Normalized confusion matrices for Study 2 using Group 1 (genes *ABCA4*, *NR2E3*, *RDH12*, *TRPM1*, *USH2A*) classifiers based on the combined FAF and pseudocolor model (a), the FAF-only model (b), and the pseudocolor-only model (c).** For each model, the columns indicate the ground truth, and the rows indicate the model's prediction.

**Experimental Study 3:** our classification of Group 2 genes based on the combination of FAF and pCF images yielded a mean accuracy of 0.85±0.03, a mean precision of 0.87±0.04, a mean F1 score of 0.83±0.04, and a mean AUC of 0.94±0.03.

The accuracy of these models was inversely correlated with the degree of confusion between classes. For the Group 1 classifiers, the combination FAF and pCF–based model and the FAF-only model, which yielded accuracy values >0.90, had a relative low rate of confusion between classes (<0.10). The most common misclassification arising from these two models was confusion between samples in gene *RDH12* being predicted as gene *USH2A*, and vice versa. Conversely, the less accurate pCF-only model had a higher rate of confusion, with predictions skewed towards the majority classes (i.e., genes *ABCA4* and *USH2A*, or classes A and Y, respectively). Nevertheless, even in this relatively less accurate model, the confusion between genes *RDH12* and *USH2A* (classes T and Y, respectively) remained, which may reflect overlapping imaging characteristics between these two genetic subtypes.

 

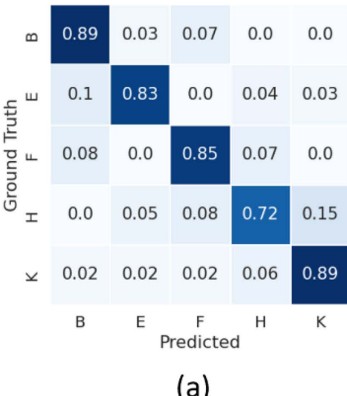
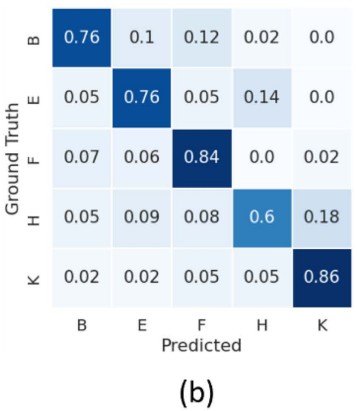
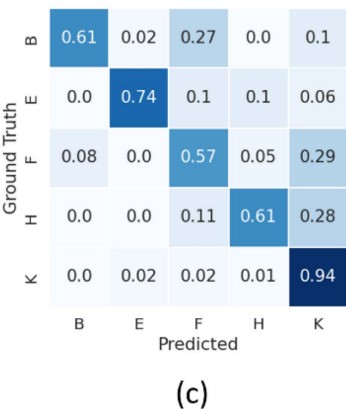

**Fig 5. Normalized confusion matrices for Study 3 using Group 2 (genes *BEST1*, *CHM*, *CNGA3*, *CRB1*, *FAM161A*) classifiers based on the combined FAF and pseudocolor model (a), the FAF-only model (b), and the pseudocolor-only model.** For each model, the columns indicate the active gene for the patient scan, and the rows indicate the model's prediction.

## 4 Discussion

In this study, we demonstrate that inherited retinal disease (IRD) can be detected with high accuracy within this retrospective single-center dataset using a deep learning model trained solely on wide-field color fundus (pCF) and fundus autofluorescence (FAF) images. These modalities are widely available in routine ophthalmic practice. Our findings show that an image-based classifier trained on genetically confirmed IRD cases can differentiate between IRD and non-IRD with high internal cross-validation performance ($0.95 \pm 0.01$), recall ($0.96 \pm 0.02$), and precision ($0.92 \pm 0.01$) (Experimental Study 1). We note that many non-IRD cases in our dataset exhibit normal-appearing FAF images, which may simplify the binary classification task. Therefore, while performance metrics are high, these results should be interpreted in the context of the dataset characteristics and internal validation design, as we did not perform a direct comparison with ophthalmologists or retinal specialists on this dataset.

A major clinical challenge in IRD care is the delay in obtaining a definitive diagnosis. Genetic testing remains the gold standard, but it is often costly, time-consuming, or inaccessible, particularly in resource-limited settings. As an example, the Eye2Gene group reported that in the UK, the diagnostic process can take over five years and cost approximately £10,000 per patient [32]. In the US, Zhang et al. [33] showed that delayed genetic testing leads to significantly higher healthcare costs ($76,838 vs. $13,084 per patient). Thus, an accurate, low-cost, and widely available image-based tool for screening IRD may have clinical relevance, although prospective validation would be required before implementation.

In Experimental Study 2, we extended our classification framework to identify the most common causative genes in our Israeli cohort. We found that FAF-based models could accurately classify patients with ABCA4 mutations (accuracy: $0.96 \pm 0.04$), likely due to the characteristic flecks and central macular involvement seen in this phenotype. NR2E3, RDH12, and FAM161A also showed strong classification performance ($\geq 0.90 \pm 0.1$). In Experimental Study 3, we tested classification of five additional IRD-associated genes and achieved a gene-group accuracy of $0.85 \pm 0.03$. Together, Group 1 and Group 2 yielded accuracy values of $0.92 \pm 0.03$ and $0.85 \pm 0.03$, respectively. We note that the gene groups (Group 1 and Group 2) were selected primarily to ensure sufficient sample sizes for robust model training rather than to reflect biological or clinical relationships. While these groups do not include all 25 genes used in Experiment 1, they enable a proof-of-concept demonstration of gene-level prediction from imaging data. Future work with larger cohorts will be needed to extend this approach to additional genes and to evaluate performance in more complex multi-gene settings.

A key finding was that multimodal input (pCF + FAF) often outperformed unimodal models. For binary IRD vs. non-IRD classification, the difference was not significant, suggesting that either modality can independently capture major distinguishing features within this dataset. However, for gene-level classification, particularly in Group 2, the combined pCF + FAF model yielded significantly higher accuracy, F1 scores, and precision compared to either modality alone. This suggests that pCF and FAF provide complementary information, FAF capturing RPE dysfunction and metabolic abnormalities, while pCF highlights vascular and structural features. The multimodal model also exhibited lower variance across performance metrics, suggesting improved robustness across cross-validation folds.

Compared to previous studies, our approach differs in study design, dataset composition, and classification scope. Prior work, such as Fujinami-Yokokawa et al. [23], used 259 FAF images to classify three genes (ABCA4, EYS, RP1L1) and achieved a mean accuracy of 0.88. Miere et al. [22] classified four disease categories using 389 FAF images, reaching an average accuracy of 0.95. While informative, these studies used only FAF and lacked comparisons to non-IRD pathologies. Shah et al. [25] used OCT data to detect Stargardt disease and achieved near-perfect performance (accuracy: 0.996), but this binary task was more constrained.

The Eye2Gene study [24,34] represents one of the most comprehensive large-scale efforts to date, including tens of thousands of images and dozens of genes. However, direct numerical comparison between their results and ours should be interpreted with caution. Their task involved classification among a much larger number of genes (up to 63), which is inherently more complex than our five-class experiments. Furthermore, their non-IRD cohort included patients with retinal and/or RPE atrophy from multiple causes, which likely increased the difficulty of binary classification compared to our dataset, where many non-IRD cases had normal or near-normal FAF imaging. Accordingly, our results should be interpreted as a targeted proof-of-concept within a targeted cohort rather than as directly comparable to large-scale multi-gene models. In our targeted five-class settings, we achieved mean accuracies of 0.92 and 0.85 for Group 1 and Group 2, respectively. These results demonstrate strong performance within a limited gene set but should not be interpreted as equivalent to large-scale multi-gene classification tasks.

Although certain IRDs, such as RP and Stargardt disease, can often be recognized through clinical examination and multimodal imaging by experienced ophthalmologists, diagnosis remains challenging in many real-world scenarios. Atypical phenotypes, early-stage disease, or overlapping features with non-IRD conditions may obscure the diagnosis. Moreover, in non-specialist or resource-limited settings where access to IRD experts and comprehensive testing is limited, deep learning-based tools may provide decision support. Such models can help general ophthalmologists detect subtle or under-recognized IRD features, enabling earlier referral and targeted genetic testing. Furthermore, the objective and reproducible nature of deep learning–based classification offers a standardized approach to initial IRD screening that is independent of examiner expertise, although its clinical impact remains to be prospectively evaluated.

While our findings demonstrate strong diagnostic performance within the studied population, they must be interpreted in the context of the genetic background on which the models were trained. The gene panel used reflects the most prevalent IRD-associated mutations in the Israeli population, and although this approach enabled robust classification within the cohort, it may not capture the full genetic and phenotypic variability seen in other populations. Therefore, the clinical applicability of our models to more diverse or globally representative cohorts remains to be determined and requires external validation.

Despite its strengths, our study has several limitations. First, although the sample size for both IRD and non-IRD groups was relatively small compared to some previous studies, our dataset is unique and of high quality, and careful patient-level partitioning together with targeted augmentation strategies helped reduce overfitting and ensured that the reported performance reflects true generalization rather than memorization of images from the same individuals within the cross validation framework. Second, the distribution of causative genes among IRD patients was uneven, with some genes represented by only a few cases. Third, we focused on only 25 of the more than 300 known IRD-associated genes, selected based on their prevalence in the Israeli population. This may limit the model's applicability to populations with

different genetic backgrounds, suggesting the need for validation in larger, more genetically diverse, and multi-center cohorts. The 25 causative genes included in our analysis accounted for approximately 63% of all IRD cases in our cohort, indicating that the gene selection reflects the most prevalent local etiologies; however, rarer genetic subtypes were not represented. Fourth, all FAF and pCF images were acquired from a single medical center using three different imaging systems, which may affect generalizability. Fifth, clinical data such as patient age, visual acuity, or disease severity were not included in the classification process; incorporating these data in the future could improve model accuracy and robustness. Sixth, no external validation was performed, and our gene groups included genes causing various retinal phenotypes; further testing on gene pairs with similar phenotypes (e.g., ABCA4 and PRPH2, DHDDS and EYS, CNGA3 and CNGB3) is needed. Seventh, the study relied on wide-field imaging systems, such as the Optos platform, which may not be available in many resource-poor or rural healthcare settings. This dependence on high-end equipment could restrict the immediate applicability of our model in such environments. To enhance accessibility, future work could focus on adapting the deep learning models to utilize images from more commonly available, lower-cost fundus cameras with narrower fields of view. Additionally, the development of AI tools compatible with portable, affordable imaging devices may facilitate broader implementation in under-resourced areas. Validation studies in these settings will be essential to assess model performance and to tailor solutions that promote equitable access to IRD diagnostics worldwide. While this study demonstrates high diagnostic accuracy of our deep learning models, it currently does not incorporate interpretability techniques such as saliency maps or Grad-CAM to explain the basis of model predictions. We acknowledge that such explainability tools are important to increase clinician trust and facilitate integration into practice. Due to resource constraints, these analyses were beyond the scope of this study but are planned for future work. In addition, the five-class gene grouping strategy simplifies the true clinical scenario in which hundreds of possible genes may be considered. While this approach was necessary due to sample size constraints, it limits direct generalization to broader diagnostic settings.

In summary, we propose a two-stage clinical workflow: (1) binary classification of IRD vs. non-IRD using multimodal images; and (2) gene-level prediction among IRD cases to guide targeted genetic testing. This framework may potentially reduce diagnostic delays, lower costs, and improve equity in IRD care. While further validation in larger, more diverse populations is needed, our study offers a proof-of-concept for integrating deep learning into clinical decision-making in inherited retinal diseases.

## Supporting information

**S1 Table. Patient data for analysis.**
(XLSX)

**S2 Table. Hyperparameters details.**
(DOCX)

## Author contributions

**Conceptualization:** Leo Joskowicz, Tim Buchbinder, Eldan Chodorov, Assaf Hoogi, Antonio Rivera, Dror Sharon, Eyal Banin, Jaime Levy.

**Data curation:** Leo Joskowicz, Tim Buchbinder, Eldan Chodorov, Assaf Hoogi, Katherine Matos, Antonio Rivera, Jaime Levy.

**Formal analysis:** Leo Joskowicz, Tim Buchbinder, Eldan Chodorov, Assaf Hoogi, Katherine Matos, Antonio Rivera, Dror Sharon, Eyal Banin, Jaime Levy.

**Investigation:** Leo Joskowicz, Jaime Levy.

**Methodology:** Leo Joskowicz, Tim Buchbinder, Eldan Chodorov, Assaf Hoogi, Katherine Matos, Antonio Rivera, Dror Sharon, Eyal Banin, Jaime Levy.

**Project administration:** Leo Joskowicz, Jaime Levy.

**Resources:** Leo Joskowicz, Katherine Matos, Antonio Rivera, Jaime Levy.

**Software:** Leo Joskowicz, Tim Buchbinder, Eldan Chodorov.

**Supervision:** Leo Joskowicz, Eldan Chodorov, Assaf Hoogi, Katherine Matos, Dror Sharon, Eyal Banin, Jaime Levy.

**Validation:** Leo Joskowicz, Tim Buchbinder, Eldan Chodorov, Assaf Hoogi, Dror Sharon, Eyal Banin, Jaime Levy.

**Visualization:** Leo Joskowicz, Tim Buchbinder, Assaf Hoogi, Antonio Rivera, Dror Sharon, Eyal Banin, Jaime Levy.

**Writing – original draft:** Leo Joskowicz, Tim Buchbinder, Eldan Chodorov, Assaf Hoogi, Katherine Matos, Antonio Rivera, Dror Sharon, Eyal Banin, Jaime Levy.

**Writing – review & editing:** Leo Joskowicz, Tim Buchbinder, Eldan Chodorov, Assaf Hoogi, Katherine Matos, Antonio Rivera, Dror Sharon, Eyal Banin, Jaime Levy.

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
