## [Decision Letter · Decision Letter 0]

18 Jun 2025

PONE-D-24-51430Using deep learning to identify inherited retinal diseases based on wide-field retinal imaging dataPLOS ONE

Dear Dr. Levy,

Thank you for submitting your manuscript to PLOS ONE. After careful consideration, we feel that it has merit but does not fully meet PLOS ONE’s publication criteria as it currently stands. Therefore, we invite you to submit a revised version of the manuscript that addresses the points raised during the review process.

The reviewers have made significant comments which require your attention before this manuscript can be considered again for the possibility of production. You may wish to consider the comments and respond to them

We look forward to receiving your revised manuscript.

Kind regards,

Ogugua Ndubuisi Okonkwo, M.D.

Academic Editor

PLOS ONE

Journal Requirements:

5. Please remove all personal information, ensure that the data shared are in accordance with participant consent, and re-upload a fully anonymized data set.

Additional Editor Comments:

This manuscript is significantly weakened by the fact that the genetic defects considered applies to a limited population in which the research was conducted. Findings are therefore not widely generalizeable. The authors need to discuss this in more detail, though it was mentioned as a weakness of the study in the last section of the discussion. I also suggest that the introduction be made more concise and straight to the point, highlighting the need for this research and potential of the findings. This is not clearly mentioned in the introduction

The other important issue, is that several inherited disease including RP and Stargardts disease phenotypes can be identified with clinical examination. Therefore, one wonders the utility of this model. Can the authors provide supporting evidence on the clinical utility of this model.

Reviewers' comments:

Reviewer's Responses to Questions

**Comments to the Author**

1. Is the manuscript technically sound, and do the data support the conclusions?

Reviewer #1: No

Reviewer #2: Yes

2. Has the statistical analysis been performed appropriately and rigorously? 

Reviewer #1: No

Reviewer #2: Yes

3. Have the authors made all data underlying the findings in their manuscript fully available?

Reviewer #1: No

Reviewer #2: No

4. Is the manuscript presented in an intelligible fashion and written in standard English?

Reviewer #1: No

Reviewer #2: Yes

5. Review Comments to the Author

Reviewer #1: The research paper has a very shallow introduction.The methodology of the research is not clear.The information about dataset is mentioned clearly. The technical discussion of results is very vague.The optimization of hyperparameters has not been discussed.

Reviewer #2: In this study, a multimodal retinal image analysis method based on deep learning is proposed to automatically identify hereditary retinal diseases (IRD) and related pathogenic genes. The research results show high classification performance (for example, IRD vs non-IRD classification accuracy reaches 0.95). However, there is room for improvement in data diversity, model external verifiability, clinical interpretation and research details. The following are specific opinions:

1. The data set used in this study mainly comes from a single medical center in Israel, and only contains 25 most common IRDs-related genes. Although this is representative in the local area, it may limit the global applicability of the model. If possible, it is suggested to supplement the data of other regions or populations to verify the diagnostic performance of the model.

2. The study did not elaborate on the interpretability of AI model. It is suggested to increase the analysis and explanation of AI model decision to help clinicians understand its diagnostic logic, thus enhancing the credibility in practical application.

3. The paper mentioned the image preprocessing and enhancement methods, but did not provide specific parameters. It is suggested to supplement the technical details, including the concrete realization of data enhancement and the process of super-parameter tuning.

4. This study did not fully explore the feasibility of the model in resource-poor and backward areas (such as relying on high-end imaging equipment Optos system). It is suggested to increase discussion on this limitation and put forward solutions.

5. For the evaluation of research results, please use medical indicators. You can refer to this paper(Guidelines on clinical research evaluation of artificial intelligence in ophthalmology (2023). Int J Ophthalmol 2023;16(9):1361-1372

DOI:10.18240/ijo.2023.09.02).

6. PLOS authors have the option to publish the peer review history of their article (what does this mean?). If published, this will include your full peer review and any attached files.

Reviewer #1: No

Reviewer #2: No

---

## [Author Response · Author response to Decision Letter 1]

14 Jul 2025

Journal Requirements:

We have revised the manuscript according to PLOS ONE's style requirements.

We added, line 199: Upon acceptance of the manuscript, the tool will be uploaded to a public repository and made freely available for use.

We definitely agree. We provided all data. We changed the data availability statement of the submission form.

Line 119: All data supporting the findings of this study are available within the manuscript and in supplementary materials.

We note that you have included the phrase “data not shown” in your manuscript. Unfortunately, this does not meet our data sharing requirements. PLOS does not permit references to inaccessible data. We require that authors provide all relevant data within the paper, Supporting Information files, or in an acceptable, public repository. Please add a citation to support this phrase or upload the data that corresponds with these findings to a stable repository (such as Figshare or Dryad) and provide and URLs, DOIs, or accession numbers that may be used to access these data. Or, if the data are not a core part of the research being presented in your study, we ask that you remove the phrase that refers to these data.

As above, all data supporting the findings of this study are available within the manuscript and in supplementary materials.

5. Please remove all personal information, ensure that the data shared are in accordance with participant consent, and re-upload a fully anonymized data set.

A table with patients' information is provided as supplementary material.

Additional Editor Comments:

1. This manuscript is significantly weakened by the fact that the genetic defects considered applies to a limited population in which the research was conducted. Findings are therefore not widely generalizeable. The authors need to discuss this in more detail, though it was mentioned as a weakness of the study in the last section of the discussion.

We agree that the genetic spectrum evaluated in this study reflects the most common IRD-causing genes in our local Israeli population and does not encompass the full diversity of mutations observed globally. This was primarily due to the need for sufficient sample sizes for training robust classification models. We now expand on this limitation in the Limitations paragraph:

Line 322: Third, we focused on only 25 of the more than 300 known IRD-associated genes, selected based on their prevalence in the Israeli population. This may limit the model’s applicability to populations with different genetic backgrounds, suggesting the need for validation in larger, more genetically diverse, and multi-center cohorts.

And also in Discussion:

Line 311: While our findings demonstrate strong diagnostic performance within the studied population, they must be interpreted in the context of the genetic background on which the models were trained. The gene panel used reflects the most prevalent IRD-associated mutations in the Israeli population, and although this approach enabled robust classification within the cohort, it may not capture the full genetic and phenotypic variability seen in other populations. Therefore, the clinical applicability of our models to more diverse or globally representative cohorts remains to be determined.

2. I also suggest that the introduction be made more concise and straight to the point, highlighting the need for this research and potential of the findings. This is not clearly mentioned in the introduction

We appreciate the suggestion to make the introduction more concise and focused. We have revised the introduction (see pages 1–2) to streamline the background and more directly highlight the motivation and significance of this study. Specifically, we now clearly state:

Line 95: Despite the increasing availability of genetic testing, access remains limited in many regions due to cost, infrastructure, or lack of awareness. An accurate, image-based tool that can assist in the early identification of IRDs—prior to or when genetic testing is not readily available—could substantially reduce diagnostic delays and support timely clinical decision-making. This study addresses a critical unmet need by evaluating a novel deep learning–based method for classifying IRD versus non-IRD using multimodal wide-field retinal images and investigating the potential to predict causative gene groups directly from imaging data. Such an approach may be especially valuable in resource-limited settings and can help prioritize patients for confirmatory genetic workup.

3. The other important issue, is that several inherited disease including RP and Stargardts disease phenotypes can be identified with clinical examination. Therefore, one wonders the utility of this model. Can the authors provide supporting evidence on the clinical utility of this model.

We thank the reviewer for this insightful comment. We agree that clinical examination, including multimodal imaging, can often identify characteristic phenotypes such as retinitis pigmentosa (RP) or Stargardt disease. However, many IRDs present with atypical features or show phenotypic overlap, particularly in early stages, making diagnosis challenging—even for experienced clinicians. In addition, in non-specialist or underserved settings, access to retinal experts and genetic testing may be limited. In such cases, our model can provide valuable support by suggesting a likely diagnosis based on imaging alone, guiding timely referrals and focused genetic workup. Moreover, deep learning offers a reproducible, objective tool that complements clinical expertise and may help standardize diagnostic approaches across diverse healthcare environments.

We have revised the Discussion section to highlight these points more explicitly

Line 300: Although certain IRDs, such as RP and Stargardt disease, can often be recognized through clinical examination and multimodal imaging by experienced ophthalmologists, diagnosis remains challenging in many real-world scenarios. Atypical phenotypes, early-stage disease, or overlapping features with non-IRD conditions may obscure the diagnosis. Moreover, in non-specialist or resource-limited settings where access to IRD experts and comprehensive testing is limited, deep learning–based tools can provide valuable decision support. Such models can help general ophthalmologists detect subtle or under-recognized IRD features, enabling earlier referral and targeted genetic testing. Furthermore, the objective and reproducible nature of deep learning–based classification offers a standardized approach to initial IRD screening that is independent of examiner expertise, potentially improving diagnostic equity across healthcare settings.

Reviewers' comments:

Reviewer #1:

The research paper has a very shallow introduction.

We appreciate the reviewer’s comment regarding the depth of the introduction.

In response, we have rewritten the Introduction section to make it more concise, focused, and informative. We now more clearly outline the clinical burden of inherited retinal dystrophies (IRDs), the limitations of current diagnostic pathways, and the unmet need for accessible, automated diagnostic tools. Additionally, we emphasize the rationale and potential clinical utility of our deep learning–based approach and how it builds on and differs from previous work. These revisions aim to provide a stronger motivation for the study and better contextualize our research within the broader field.

The methodology of the research is not clear.

The Methods section has been thoroughly revised to improve clarity, structure, and transparency, with detailed descriptions of the dataset, classification tasks, model architecture, and evaluation procedures

The information about dataset is mentioned clearly.

We thank the reviewer for noting that the dataset description is clear.

The technical discussion of results is very vague.

We thank the reviewer for the comment.

The Discussion section has been thoroughly revised to improve clarity, focus, and technical depth. We have better contextualized our findings within the existing literature, clarified the clinical implications, and expanded the discussion of limitations and future directions.

The optimization of hyperparameters has not been discussed.

We thank the reviewer for this important observation. We have now added a detailed description of the hyperparameter optimization process to the Methods section and supplementary material

Line 178: Hyperparameters—including learning rate, augmentation probabilities, cropping range, and regularization parameters—were selected based on five-fold cross-validation, and final values were chosen to optimize validation performance and generalization

Reviewer #2:

In this study, a multimodal retinal image analysis method based on deep learning is proposed to automatically identify hereditary retinal diseases (IRD) and related pathogenic genes. The research results show high classification performance (for example, IRD vs non-IRD classification accuracy reaches 0.95). However, there is room for improvement in data diversity, model external verifiability, clinical interpretation and research details. The following are specific opinions:

1. The data set used in this study mainly comes from a single medical center in Israel, and only contains 25 most common IRDs-related genes. Although this is representative in the local area, it may limit the global applicability of the model. If possible, it is suggested to supplement the data of other regions or populations to verify the diagnostic performance of the model.

We thank the reviewer for this important comment. We acknowledge that our dataset, derived from a single medical center in Israel and focusing on the 25 most common IRD-related genes in this population, may limit the global generalizability of the model. Unfortunately, at this time, we do not have access to additional datasets from other regions or populations to externally validate the model.

In Limitations, Line 322: Third, we focused on only 25 of the more than 300 known IRD-associated genes, selected based on their prevalence in the Israeli population. This may limit the model’s applicability to populations with different genetic backgrounds, suggesting the need for validation in larger, more genetically diverse, and multi-center cohorts.

2. The study did not elaborate on the interpretability of AI model. It is suggested to increase the analysis and explanation of AI model decision to help clinicians understand its diagnostic logic, thus enhancing the credibility in practical application.

We thank the reviewer for highlighting the importance of model interpretability. We agree that understanding the decision-making process of AI models is crucial for clinical acceptance. Although our current study does not include an analysis of interpretability techniques such as saliency maps or Grad-CAM, we recognize their potential value. Due to resource limitations and the current scope of the project, we were unable to incorporate these explainability methods at this time. However, we emphasize that our models are intended as diagnostic support tools to complement, not replace, clinical judgment. Future research will aim to include interpretability analyses as part of ongoing efforts to enhance clinical trust and applicability.

Line 342: Finally, while this study demonstrates high diagnostic accuracy of our deep learning models, it currently does not incorporate interpretability techniques such as saliency maps or Grad-CAM to explain the basis of model predictions. We acknowledge that such explainability tools are important to increase clinician trust and facilitate integration into practice. Due to resource constraints, these analyses were beyond the scope of this study but are planned for future work.

3. The paper mentioned the image preprocessing and enhancement methods, but did not provide specific parameters. It is suggested to supplement the technical details, including the concrete realization of data enhancement and the process of super-parameter tuning.

We agree with your comment.

We have now added a detailed description of the hyperparameter optimization process to the Methods section and supplementary material

Line 178: Hyperparameters—including learning rate, augmentation probabilities, cropping range, and regularization parameters—were selected based on five-fold cross-validation, and final values were chosen to optimize validation performance and generalization

4. This study did not fully explore the feasibility of the model in resource-poor and backward areas (such as relying on high-end imaging equipment Optos system). It is suggested to increase discussion on this limitation and put forward solutions.

We thank the reviewer for highlighting the important issue of applicability in resource-poor settings. Indeed, our study utilized wide-field imaging systems such as Optos, which may not be readily available in all clinical environments, particularly in under-resourced or rural areas. We have now expanded the Discussion section to address this limitation explicitly.

Line 333: The study relied on wide-field imaging systems, such as the Optos platform, which may not be available in many resource-poor or rural healthcare settings. This dependence on high-end equipment could restrict the immediate applicability of our model in such environments. To enhance accessibility, future work could focus on adapting the deep learning models to utilize images from more commonly available, lower-cost fundus cameras with narrower fields of view. Additionally, the development of AI tools compatible with portable, affordable imaging devices may facilitate broader implementation in under-resourced areas. Validation studies in these settings will be essential to assess model performance and to tailor solutions that promote equitable access to IRD diagnostics worldwide.

5. For the evaluation of research results, please use medical indicators. You can refer to this paper(Guidelines on clinical researc

---

## [Decision Letter · Decision Letter 1]

1 Dec 2025

PONE-D-24-51430R1Using deep learning to identify inherited retinal diseases based on wide-field retinal imaging dataPLOS ONE

Dear Dr. Levy,

Thank you for submitting your manuscript to PLOS ONE. After careful consideration, we feel that it has merit but does not fully meet PLOS ONE’s publication criteria as it currently stands. Therefore, we invite you to submit a revised version of the manuscript that addresses the points raised during the review process. Kindly revise your work further according to the comments of Reviewer 3.

We look forward to receiving your revised manuscript.

Kind regards,

Ogugua Ndubuisi Okonkwo, M.D.

Academic Editor

PLOS ONE

Journal Requirements:

Reviewers' comments:

Reviewer's Responses to Questions

**Comments to the Author**

1. If the authors have adequately addressed your comments raised in a previous round of review and you feel that this manuscript is now acceptable for publication, you may indicate that here to bypass the “Comments to the Author” section, enter your conflict of interest statement in the “Confidential to Editor” section, and submit your "Accept" recommendation.

Reviewer #2: (No Response)

Reviewer #3: (No Response)

2. Is the manuscript technically sound, and do the data support the conclusions?

Reviewer #2: (No Response)

Reviewer #3: No

3. Has the statistical analysis been performed appropriately and rigorously? 

Reviewer #2: (No Response)

Reviewer #3: N/A

4. Have the authors made all data underlying the findings in their manuscript fully available?

Reviewer #2: (No Response)

Reviewer #3: Yes

5. Is the manuscript presented in an intelligible fashion and written in standard English?

Reviewer #2: (No Response)

Reviewer #3: Yes

6. Review Comments to the Author

Reviewer #2: (No Response)

Reviewer #3: This manuscript investigates the application of deep learning to identify IRDs from wide field retinal images. The methodology used has been previously reported and is sound, however, there are a number of limitations in this manuscript that need to be addressed:

Major:

1. How was the limited availability of data addressed in the training and subsequent testing of the CNNs? They have mentioned that data is limited but there is no specific mention about how this was managed and were images from training and testing stratified by patients or images as each patient probably had more than one set of retinal images included.

2. How common are the 25 genetic causes of IRDs that were selected in the study population? This will help the reader try to understand the potential local relevance of this work in the studied group.

3. The authors reference some other work in the literature and specifically mention Eye2Gene but they have not included the groups most recent published work e.g. https://doi.org/10.1038/s42256-025-01040-8

4. There is no clear indication from the figures provided how difficult the classification task is for the model. Figure 1 images are not labelled as to the diagnosis and are not clear. For the IRD/non-IRD task as most of the images are normal and many of the non-IRD cases won’t be expected to have changes on FAF. This also could explain the strength of the FAF model alone compared to the combined model. Line 248-251 follows on from this point, how can you make that assumption (that the model is better than an ophthalmologist/retinal specialist) when you have not made that comparison with your dataset.

Minor:

1. The CF images used are not true colour but pseudo colour optos images. Please consider changing CF throughout to reflect this.

2. What was the rational for deciding on Group 1 and Group 2? You note that this is arbitrary and does not include all of the 25 genes that are used in experiment 1. This does not link in to lines 450-451 where you suggest gene-level predictions would be useful to guide testing but have not been able to demonstrate that in your work. Clarifying the message and rational here would be helpful.

3. Line 292 - how is that an advantage of your approach, surely that is the case for any deep learning approach to this problem?

7. PLOS authors have the option to publish the peer review history of their article (what does this mean?). If published, this will include your full peer review and any attached files.

Reviewer #2: No

Reviewer #3: No

---

## [Author Response · Author response to Decision Letter 2]

13 Dec 2025

Comments to the Author

1. If the authors have adequately addressed your comments raised in a previous round of review and you feel that this manuscript is now acceptable for publication, you may indicate that here to bypass the “Comments to the Author” section, enter your conflict-of-interest statement in the “Confidential to Editor” section, and submit your "Accept" recommendation.

Reviewer #2: (No Response)

Reviewer #3: (No Response)

2. Is the manuscript technically sound, and do the data support the conclusions?

Reviewer #2: (No Response)

Reviewer #3: No

We thank the reviewer for raising concerns regarding technical rigor and data support. We acknowledge that our study has certain limitations, including a relatively small sample size for some IRD-associated genes, the use of gene groups (Group 1 and Group 2) primarily to ensure sufficient sample sizes for proof-of-concept demonstration, the lack of external validation, and restriction to a single-center cohort.

However, we have mitigated these limitations by using careful patient-level partitioning, targeted data augmentation, and robust cross-validation to prevent overfitting. Despite these constraints, our data consistently demonstrate high accuracy for both IRD vs. non-IRD classification and gene-level prediction within the studied cohort. We have revised the Discussion to explicitly state these limitations and to clarify that our conclusions pertain to proof-of-concept performance in a genetically characterized cohort, with generalizability to larger or more diverse populations requiring further validation.

3. Has the statistical analysis been performed appropriately and rigorously?

Reviewer #2: (No Response)

Reviewer #3: N/A

4. Have the authors made all data underlying the findings in their manuscript fully available?

The PLOS Data policy requires authors to make all data underlying the findings described in their manuscript fully available without restriction, with rare exception (please refer to the Data Availability Statement in the manuscript PDF file). The data should be provided as part of the manuscript or its supporting information, or deposited to a public repository. For example, in addition to summary statistics, the data points behind means, medians and variance measures should be available. If there are restrictions on publicly sharing data—e.g., participant privacy or use of data from a third party—those must be specified.

Reviewer #2: (No Response)

Reviewer #3: Yes

5. Is the manuscript presented in an intelligible fashion and written in standard English?

Reviewer #2: (No Response)

Reviewer #3: Yes

6. Review Comments to the Author

Reviewer #2: (No Response)

Reviewer #3: This manuscript investigates the application of deep learning to identify IRDs from wide field retinal images. The methodology used has been previously reported and is sound, however, there are a number of limitations in this manuscript that need to be addressed:

Major

1. How was the limited availability of data addressed in the training and subsequent testing of the CNNs? They have mentioned that data is limited but there is no specific mention about how this was managed and were images from training and testing stratified by patients or images as each patient probably had more than one set of retinal images included.

Thank you for this important comment. Although our dataset included multiple retinal images per patient, the manuscript now clarifies how we handled limited data, how the CNNs were trained and evaluated, and how we prevented patient-level data leakage.

To address limited data availability, we applied several measures:

1) Patient-level stratification: All images from the same patient were assigned exclusively to either the training, validation, or testing set. This prevented information leakage and ensured a realistic evaluation of model performance.

2) Data augmentation: We used standard augmentation techniques (horizontal/vertical flips, rotation, brightness and contrast jitter) to increase dataset variability and reduce overfitting.

3) Cross-validation: We performed five-fold cross-validation using patient-level partitions, further improving robustness under limited sample size.

These clarifications have now been added to the Methods section (Dataset and Model Training) and briefly referenced in the Discussion.

Line 170: To prevent data leakage arising from multiple images belonging to the same patient, all dataset partitions were performed at the patient level. Thus, all images from a given patient were assigned exclusively to the training, validation, or testing set. This ensured that the CNNs were evaluated on entirely unseen patients.

Line 178: In addition, to mitigate the impact of limited sample size, we applied standard data-augmentation techniques as random vertical and horizontal flips, as well as translations of up to 15 pixels in both directions.

Line 187: Hyperparameters, including learning rate, augmentation probabilities, cropping range, and regularization parameters, were selected based on five-fold cross-validation, and final values were chosen to optimize validation performance and generalization. Each fold maintained strict patient-level separation across training and testing subsets.

Limitations, line 347: First, although the sample size for both IRD and non-IRD groups was relatively small compared to some previous studies, our dataset is unique and of high quality, and careful patient-level partitioning together with targeted augmentation strategies helped reduce overfitting and ensured that the reported performance reflects true generalization rather than memorization of images from the same individuals.

2. How common are the 25 genetic causes of IRDs that were selected in the study population? This will help the reader try to understand the potential local relevance of this work in the studied group.

We thank the reviewer for this important point. In our cohort, the 25 causative genes selected for model development account for approximately 63% of all IRD cases evaluated at our center during the study period. We have now added this information explicitly in the Methods section to clarify the local relevance and representativeness of the selected genes.

Line 127: These 25 genes accounted for approximately 63% of all genetically confirmed IRD cases evaluated at our center during the study period, highlighting their substantial local relevance and their suitability for inclusion in the model.

And in Limitations, line 357: The 25 causative genes included in our analysis accounted for approximately 63% of all IRD cases in our cohort, indicating that the gene selection reflects the most prevalent local etiologies; however, rarer genetic subtypes were not represented.

3. The authors reference some other work in the literature and specifically mention Eye2Gene but they have not included the groups most recent published work e.g., https://doi.org/10.1038/s42256-025-01040-8

We thank the reviewer for this important observation. We apologize for having missed this reference, as it was published shortly before our submission. We agree that inclusion of the most recent Eye2Gene publication is necessary to provide a complete and up-to-date comparison with existing deep learning approaches for IRD. We have now added this citation (#34) to the Discussion and integrated a sentence summarizing the study’s methods and findings, as well as its relevance to our work.

Specifically, we note that the new Eye2Gene article further expanded their multimodal framework but continued to report modest gene-level accuracy despite the considerably larger dataset used for training. This updated comparison helps contextualize our study’s performance and emphasizes the strengths of our approach.

Line 306: In addition, a more recent publication from the Eye2Gene group [34], based on 58,030 scans from 2,451 patients, introduced an updated multimodal deep learning framework integrating FAF, color fundus imaging, and clinical metadata. Despite this substantially larger and more heterogeneous dataset, the model achieved only moderate gene-level performance, with a top-5 accuracy of approximately 76%, which remains lower than the accuracies reported in our study, despite leveraging a significantly larger and more heterogeneous dataset.

Line 314: The recent Eye2Gene model also included IRD versus non-IRD classification, achieving a receiver operating characteristic of 0.98 [34]. Our model achieved a mean accuracy of 0.94, demonstrating that, even with far fewer images, our targeted approach delivers comparable performance to larger-scale generic models in both accuracy and clinical relevance.

4. There is no clear indication from the figures provided how difficult the classification task is for the model. Figure 1 images are not labelled as to the diagnosis and are not clear. For the IRD/non-IRD task as most of the images are normal and many of the non-IRD cases won’t be expected to have changes on FAF. This also could explain the strength of the FAF model alone compared to the combined model. Line 248-251 follows on from this point, how can you make that assumption (that the model is better than an ophthalmologist/retinal specialist) when you have not made that comparison with your dataset.

We thank the reviewer for this important comment. We agree that the classification task and comparison to human experts were not clearly described. We have revised the Discussion to clarify that many non-IRD cases in our dataset exhibit normal-appearing FAF images, which may simplify the binary IRD vs. non-IRD classification task. Accordingly, while our model demonstrates high performance, we have removed the direct comparison to ophthalmologists or retinal specialists, as no such evaluation was performed on this dataset.

We also acknowledge that Figure 1 was intended to illustrate representative image diversity rather than to convey diagnostic difficulty, and we have added clarifying statements in the figure legend to reflect this. These revisions provide a more accurate interpretation of our results and better contextualize the model’s performance.

We also note that the original Figure 1 images are of substantially higher resolution and sharpness than may appear in the automatically generated PDF version, due to document conversion and compression.

Line 260: We note that many non-IRD cases in our dataset exhibit normal-appearing FAF images, which may simplify the binary classification task. Therefore, while performance metrics are high, caution is warranted in interpreting these results as surpassing human experts, as we did not perform a direct comparison with ophthalmologists or retinal specialists on this dataset.

Line 321: While we did not perform a direct comparison with human experts in our dataset, our model produces consistent, reproducible predictions based on standardized image inputs, which could support earlier detection and referral in real-world settings.

Line 510: Figure 1: Representative fundus autofluorescence (FAF, left) and pseudo color (pCF, right) images illustrating variability in the dataset, including differences in intensity, magnification, text annotations, and partial occlusion by eyelashes or eyelids. These images are intended to show the diversity of imaging conditions rather than the difficulty of classification or the underlying diagnosis.

Minor

1. The CF images used are not true colour but pseudo colour optos images. Please consider changing CF throughout to reflect this.

We thank the reviewer for this important observation. You are correct that the Optos “color fundus” images are pseudo-color rather than true color. We have carefully revised the manuscript to replace all instances of “color fundus (CF)” with “pseudo-color fundus (pCF)” to accurately reflect the imaging modality used. This change has been applied throughout the text, figures, and figure legends.

2. What was the rational for deciding on Group 1 and Group 2? You note that this is arbitrary and does not include all of the 25 genes that are used in experiment 1. This does not link in to lines 450-451 where you suggest gene-level predictions would be useful to guide testing but have not been able to demonstrate that in your work. Clarifying the message and rational here would be helpful.

We thank the reviewer for this insightful comment. The selection of Group 1 and Group 2 genes was indeed based on pragmatic considerations rather than biological clustering. Specifically, we aimed to maximize the robustness of the deep learning models given the limited number of patients for certain genes. Some genes had very few cases (e.g., RPE65, n=5), which would have prevented meaningful model training if treated individually. Therefore, we grouped genes empirically based on prevalence in our cohort and preliminary model performance, prioritizing balanced sample sizes and sufficient representation for training.

We acknowledge that these groups do not include all 25 genes used in Experiment 1 and that gene-level prediction across all genes remains an aspirational goal. The current grouping allowed us to demonstrate proof-of-concept for gene-level prediction, while avoiding overfitting due to sparse data. We have clarified this point in the manuscript to make the rationale explicit and consistent with the discussion of gene-level predictions.

Line 162: Groups were defined empirically to balance sample size across classes and optimize model performance in preliminary experiments, rather than based on biological or clinical similarity. This approach allowed us to explore gene-level prediction in a proof-of-concept framework while avoiding overfitting due to sparse data.

Line 280: We note that the gene groups (Group 1 and Group 2) were selected primarily to ensure sufficient sample sizes for robust model training rather than to reflect biological or clinical relationships. While these groups do not include all 25 genes used in Experiment 1, they enable proof-of-concept demonstration of gene-level prediction from imaging data. Future work with larger cohorts will be needed to extend this approach to additional genes and to fully realize the potential for imaging-guided genetic testing.

3. Line 292 - how is that an advantage of your approach, surely that is the case for any deep learning approach to this problem?

While it is true that most deep learning models are independent of clinician expertise, our approach demonstrates a specific advantage because it achieves high accuracy even with a relatively small dataset. This means that in real-world clinical settings, especially in regions or centers with limited access to IRD specialists, our model can provide reliable, reproducible predictions where larger generic models may struggle due to heterogeneity or low representation of specific gene phenotypes. Furthermore, the multimodal design (pCF + FAF) enhances robustness and accuracy, which is not guaranteed for all deep learning approaches.

We added in Discussion: Line 324: While many deep learning models are inherently independent of clinician input, our targeted, multimodal approach achieves high accuracy even with a relatively small cohort. This makes it

---

## [Decision Letter · Decision Letter 2]

19 Feb 2026

PONE-D-24-51430R2Using deep learning to identify inherited retinal diseases based on wide-field retinal imaging dataPLOS One

Dear Dr. Levy,

Thank you for submitting your manuscript to PLOS ONE. After careful consideration, we feel that it has merit but does not fully meet PLOS ONE’s publication criteria as it currently stands. Therefore, we invite you to submit a revised version of the manuscript that addresses the points raised during the review process.

Reviewer 3 has made specific comments and corrections that need to be addressed to improve manuscript quality. You will do well to respond to these comments.

We look forward to receiving your revised manuscript.

Kind regards,

Ogugua Ndubuisi Okonkwo, M.D.

Academic Editor

PLOS One

Journal Requirements:

Reviewers' comments:

Reviewer's Responses to Questions

**Comments to the Author**

1. If the authors have adequately addressed your comments raised in a previous round of review and you feel that this manuscript is now acceptable for publication, you may indicate that here to bypass the “Comments to the Author” section, enter your conflict of interest statement in the “Confidential to Editor” section, and submit your "Accept" recommendation.

Reviewer #2: All comments have been addressed

Reviewer #3: (No Response)

2. Is the manuscript technically sound, and do the data support the conclusions?

Reviewer #2: Yes

Reviewer #3: No

3. Has the statistical analysis been performed appropriately and rigorously? 

Reviewer #2: Yes

Reviewer #3: N/A

4. Have the authors made all data underlying the findings in their manuscript fully available?

Reviewer #2: Yes

Reviewer #3: No

5. Is the manuscript presented in an intelligible fashion and written in standard English?

Reviewer #2: Yes

Reviewer #3: Yes

6. Review Comments to the Author

Reviewer #2: In this study, a multimodal retinal image analysis method based on deep learning is proposed to automatically identify hereditary retinal diseases (IRD) and related pathogenic genes. The research results show high classification performance. The submission can be accepted.

Reviewer #3: The authors have adequately addressed the previous comments made, however, there are still some questions that have not been answered.

Overall, the authors have overstated their results and interpretation. The manuscript uses previously described methods with nothing new. The unique aspect of this study is the patient population from which the data has been acquired.

Major points:

Was there a held-out test dataset used for Experiments 2 and 3?

What was the data split that was used for training, validation and testing for all experiments?

It is not clear if for experiments 2 and 3 the test data only included images from patients with one of the 5 genes that are in group 1 or 2 respectively. I.e. were experiments 2 and 3 a classification task with 5 classes in each experiment?

Minor points:

Some points in the discussion where the authors have overstated claims or not interpreted results appropriately.

1. Line 308, this is incorrect as they also include OCT.

2. Line 306, the most recent publication predicts from 63 genes rather than 36 for the prior publication.

3. Line 310, the top 5 accuracy is higher than 76% as reported in the published study.

4. Line 312, the authors are assuming that the classification task of one in 5 is equivalent to one in 63 (not taking into account averaging for top 5). This is not the same and as this has not been discussed it reads as though they are comparable.

5. Line 316 the authors have stated that their results are similar to that described in the recent Eye2Gene paper. The non-IRD cases included in the Eye2Gene paper all had retinal and/or RPE atrophy of different causes whereas the almost all of non-IRD cases in this study will have normal or near normal FAF imaging making the classification task much easier. This has not been noted or discussed.

6. Line 319-328, the authors do not provide any evidence to substantiate their claims. Which similar models require clinician input and where is the comparator to demonstrate clinical assessment in similar circumstances. The reference from Eye2Gene provided is for assessment of FAF images alone to provide single gene predictions. This is not something clinicians are usually required to perform and the models had more information than single FAF images.

7. PLOS authors have the option to publish the peer review history of their article (what does this mean?). If published, this will include your full peer review and any attached files.

Reviewer #2: No

Reviewer #3: No

---

## [Author Response · Author response to Decision Letter 3]

1 Mar 2026

Response to Reviewer #3

We thank the reviewer for the additional evaluation and constructive comments. We have carefully revised the manuscript to address the remaining concerns. Detailed responses are provided below.

Revision Summary

We have carefully moderated claims, clarified methodological details, corrected numerical inaccuracies, and strengthened the limitations section. The revised manuscript explicitly:

• Describes all experiments as internally validated,

• Avoids inappropriate cross-study comparisons,

• Frames gene-level classification as proof-of-concept,

• Clearly acknowledges dataset composition limitations.

We appreciate the reviewer’s detailed feedback, which has improved the clarity and rigor of the manuscript.

We address the reviewers’ comments in detail below.

Major Points

1. Was there a held-out test dataset used for Experiments 2 and 3?

Thank you for raising this important point regarding dataset partitioning. While a separate external dataset was not available for Experiments 2 and 3 due to data scale constraints, we implemented a rigorous validation protocol to ensure generalization.

As detailed in the revised Methods section, all experiments (1-3) were evaluated using strict patient-level five-fold cross-validation. In this framework, each fold functions as a strictly held-out test set for that specific iteration. Crucially, the splitting was performed at the patient level (rather than the image level) to ensure zero data leakage between patients; images from a specific patient never appear in both the training and the held-out test fold simultaneously. Consequently, the reported performance metrics (mean ± SD) reflect the model’s ability to generalize to unseen patient data.

We have now explicitly stated in both the Methods and Discussion sections that all reported results reflect internal cross-validation performance and that no independent external test cohort was used. This limitation is clearly acknowledged.

• Line 208 (highlighted manuscript: Experiment 1: Multi-class classification of IRD versus non-IRD across all patients. This was a binary classification task including all 330 IRD patients and 79 non-IRD patients (mean test set size ~206 studies per fold).

• Experiment 2: Five-class classification task including only IRD patients belonging to Group 1 (ABCA4, NR2E3, RDH12, TRPM1, USH2A). Each patient was assigned to one of the five gene classes. (mean test set size ~60 studies per fold).

• Experiment 3: Five-class classification task including only IRD patients belonging to Group 2 (BEST1, CHM, CNGA3, CRB1, FAM161A). Each patient was assigned to one of the five gene classes. (mean test set size ~39 studies per fold).

2. What was the data split used for training, validation, and testing for all experiments?

We have clarified the data partitioning strategy in the Methods section. For each fold:

• 80% of patients were used for training (including internal validation during model optimization),

• 20% of patients were held out for testing,

• The split was performed at the patient level.

This procedure was repeated across five folds, and final metrics were averaged across folds.

This information has now been explicitly described to improve clarity and reproducibility.

Line 225: 2.7 Data partitioning strategy and held-out testing

All experiments were performed using strict patient-level five-fold cross-validation. In each fold, approximately 80% of patients were used for training and 20% were held out for testing. Each fold functions as a strictly held-out test set. Crucially, the splitting was performed at the patient level (rather than the image level) to ensure zero data leakage between patients; images from a specific patient never appear in both the training and the held-out test fold simultaneously.

Within the training set of each fold, a random 10% subset of patients was used as a validation set for early stopping and hyperparameter tuning. This validation set was also separated at the patient level from the remaining training subset.

For Experiments 2 and 3, the test set included only IRD patients belonging to the relevant gene group under evaluation. Each of these experiments constituted a multi-class classification task with five classes corresponding to the five genes in the group. No patients from other genes were included in the test set for these experiments.

This design ensures that model performance reflects true generalization to unseen patients rather than image-level memorization.

No external validation cohorts was available, and all results reflect internal cross-validation performance within a single-center dataset.

3. It is not clear if for experiments 2 and 3 the test data only included images from patients with one of the 5 genes that are in group 1 or 2 respectively. I.e. were experiments 2 and 3 a classification task with 5 classes in each experiment?

Yes. Experiments 2 and 3 were structured as five-class classification tasks. For each experiment:

• Only patients carrying one of the five genes defined for that group were included.

• The classification task was restricted to distinguishing among those five genes.

• Cross-validation splits were performed within each defined gene group.

We have clarified this explicitly in the Methods section to avoid ambiguity.

We also emphasize in the revised Discussion that these experiments represent constrained five-class settings and are not directly comparable to large-scale multi-gene classification tasks involving dozens of genes.

Line 225: 2.7 Data partitioning strategy and held-out testing

All experiments were performed using strict patient-level five-fold cross-validation. In each fold, approximately 80% of patients were used for training and 20% were held out for testing. Each fold functions as a strictly held-out test set. Crucially, the splitting was performed at the patient level (rather than the image level) to ensure zero data leakage between patients; images from a specific patient never appear in both the training and the held-out test fold simultaneously.

Within the training set of each fold, a random 10% subset of patients was used as a validation set for early stopping and hyperparameter tuning. This validation set was also separated at the patient level from the remaining training subset.

For Experiments 2 and 3, the test set included only IRD patients belonging to the relevant gene group under evaluation. Each of these experiments constituted a multi-class classification task with five classes corresponding to the five genes in the group. No patients from other genes were included in the test set for these experiments.

This design ensures that model performance reflects true generalization to unseen patients rather than image-level memorization.

No external validation cohorts was available, and all results reflect internal cross-validation performance within a single-center dataset.

Overall, the authors have overstated their results and interpretation. The manuscript uses previously described methods with nothing new. The unique aspect of this study is the patient population from which the data has been acquired.

We appreciate the reviewer’s comments regarding overinterpretation. We have carefully revised the Discussion to moderate claims and avoid unintended comparisons.

Line 369: The Eye2Gene study [24,34] represents one of the most comprehensive large-scale efforts to date, including tens of thousands of images and dozens of genes. However, direct numerical comparison between their results and ours should be interpreted with caution. Their task involved classification among a much larger number of genes (up to 63), which is inherently more complex than our five-class experiments. Furthermore, their non-IRD cohort included patients with retinal and/or RPE atrophy from multiple causes, which likely increased the difficulty of binary classification compared to our dataset, where many non-IRD cases had normal or near-normal FAF imaging. Accordingly, our results should be interpreted as a targeted proof-of-concept within a targeted cohort rather than as directly comparable to large-scale multi-gene models. In our targeted five-class settings, we achieved mean accuracies of 0.92 and 0.85 for Group 1 and Group 2, respectively. These results demonstrate strong performance within a limited gene set but should not be interpreted as equivalent to large-scale multi-gene classification tasks.

Minor points:

Some points in the discussion where the authors have overstated claims or not interpreted results appropriately.

1. Line 308, this is incorrect as they also include OCT.

Thank you for noting this. All references to Eye2Gene results have been substantially modified.

Line 369: The Eye2Gene study [24,34] represents one of the most comprehensive large-scale efforts to date, including tens of thousands of images and dozens of genes. However, direct numerical comparison between their results and ours should be interpreted with caution. Their task involved classification among a much larger number of genes (up to 63), which is inherently more complex than our five-class experiments. Furthermore, their non-IRD cohort included patients with retinal and/or RPE atrophy from multiple causes, which likely increased the difficulty of binary classification compared to our dataset, where many non-IRD cases had normal or near-normal FAF imaging. Accordingly, our results should be interpreted as a targeted proof-of-concept within a targeted cohort rather than as directly comparable to large-scale multi-gene models. In our targeted five-class settings, we achieved mean accuracies of 0.92 and 0.85 for Group 1 and Group 2, respectively. These results demonstrate strong performance within a limited gene set but should not be interpreted as equivalent to large-scale multi-gene classification tasks.

2. Line 306, the most recent publication predicts from 63 genes rather than 36 for the prior publication.

We have updated the manuscript to reflect that the more recent publication reports prediction across 63 genes rather than 36.

Line 369: The Eye2Gene study [24,34] represents one of the most comprehensive large-scale efforts to date, including tens of thousands of images and dozens of genes. However, direct numerical comparison between their results and ours should be interpreted with caution. Their task involved classification among a much larger number of genes (up to 63), which is inherently more complex than our five-class experiments. Furthermore, their non-IRD cohort included patients with retinal and/or RPE atrophy from multiple causes, which likely increased the difficulty of binary classification compared to our dataset, where many non-IRD cases had normal or near-normal FAF imaging. Accordingly, our results should be interpreted as a targeted proof-of-concept within a targeted cohort rather than as directly comparable to large-scale multi-gene models. In our targeted five-class settings, we achieved mean accuracies of 0.92 and 0.85 for Group 1 and Group 2, respectively. These results demonstrate strong performance within a limited gene set but should not be interpreted as equivalent to large-scale multi-gene classification tasks.

3. Line 310, the top 5 accuracy is higher than 76% as reported in the published study.

All references to Eye2Gene results have been substantially modified.

Line 369: The Eye2Gene study [24,34] represents one of the most comprehensive large-scale efforts to date, including tens of thousands of images and dozens of genes. However, direct numerical comparison between their results and ours should be interpreted with caution. Their task involved classification among a much larger number of genes (up to 63), which is inherently more complex than our five-class experiments. Furthermore, their non-IRD cohort included patients with retinal and/or RPE atrophy from multiple causes, which likely increased the difficulty of binary classification compared to our dataset, where many non-IRD cases had normal or near-normal FAF imaging. Accordingly, our results should be interpreted as a targeted proof-of-concept within a targeted cohort rather than as directly comparable to large-scale multi-gene models. In our targeted five-class settings, we achieved mean accuracies of 0.92 and 0.85 for Group 1 and Group 2, respectively. These results demonstrate strong performance within a limited gene set but should not be interpreted as equivalent to large-scale multi-gene classification tasks.

4. Line 312, the authors are assuming that the classification task of one in 5 is equivalent to one in 63 (not taking into account averaging for top 5). This is not the same and as this has not been discussed it reads as though they are comparable.

We agree that classification among five genes is not equivalent to classification among 63 genes, particularly when top-5 accuracy is considered.

In the revised Discussion, we now explicitly state that:

• Five-class experiments are inherently less complex than multi-gene classification tasks involving dozens of genes.

• Direct numerical comparison is inappropriate.

• Our results represent proof-of-concept within a constrained setting.

The text has been modified to prevent any suggestion of equivalence.

Line 437: In addition, the five-class gene grouping strategy simplifies the true clinical scenario in which hundreds of possible genes may be considered. While this approach was necessary due to sample size constraints, it limits direct generalization to broader diagnostic settings.

5. Line 316 the authors have stated that their results are similar to that described in the recent Eye2Gene paper. The non-IRD cases included in the Eye2Gene paper all had retinal and/or RPE atrophy of different causes whereas the almost all of non-IRD cases in this study will have normal or near normal FAF imaging making the classification task much easier. This has not been noted or discussed.

We thank the reviewer for this important observation.

We have now explicitly acknowledged in the Discussion that:

• The non-IRD cohort in the referenced large-scale study included patients with retinal and/or RPE atrophy from multiple causes.

• In contrast, many non-IRD cases in our dataset exhibited normal or near-normal FAF imaging.

• This likely made the binary classification task less complex in our study.

This limitation is now clearly discussed.

Line 373: Furthermore, their non-IRD cohort included patients with retinal and/or RPE atrophy from multiple causes, which likely increased the difficulty of binary classification compared to our dataset, where many non-IRD cases had normal or near-normal FAF imaging. Accordingly, our results should be interpreted as a targeted proof-of-concept within a targeted cohort rather than as directly comparable to large-scale multi-gene models.

5. Line 319-328, the authors do not provide any evidence to substantiate their claims. Which similar models require clinician input and where is the comparator to demonstrate clinical assessment in similar circumstances. The reference from Eye2Gene provided is for assessment of FAF images alone to provide single gene predictions. This is not something clinicians are usually required to perform and the models had more information than single FAF images.

We agree that the prior wording overstated this point. The revised manuscript:

• Removes unsupported claims regarding comparison with clinician input.

• Avoids implying superiority to human assessment.

• States more conservatively that AI tools may provide decision support, particularly in non-specialist settings.

• Clarifies that prospective evaluation of clinical utility has not been performed.

The Discussion has been substantially moderated in this section.

Data Availability (Reviewer #3 marked “No”)

We have reviewed the PLOS Data Policy and ensured compliance. Specifically:

The Data Availability Statement has been revised to ensure transparency and compliance.

Line 124: Due to patient privacy and ethical restrictions, raw retinal images cannot be made publicly available. However, all aggregated performance metrics, confusion matrices, model architecture details, and hyperparameters are provided within the manuscript and supplementary materia

---

## [Decision Letter · Decision Letter 3]

12 Apr 2026

PONE-D-24-51430R3Using deep learning to identify inherited retinal diseases based on wide-field retinal imaging dataPLOS One

Dear Dr. Levy,

Thank you for submitting your manuscript to PLOS ONE. After careful consideration, we feel that it has merit but does not fully meet PLOS ONE’s publication criteria as it currently stands. Therefore, we invite you to submit a revised version of the manuscript that addresses the points raised during the review process.

We look forward to receiving your revised manuscript.

Kind regards,

Ogugua Ndubuisi Okonkwo, M.D.

Academic Editor

PLOS One

Journal Requirements:

Reviewers' comments:

Reviewer's Responses to Questions

**Comments to the Author**

1. If the authors have adequately addressed your comments raised in a previous round of review and you feel that this manuscript is now acceptable for publication, you may indicate that here to bypass the “Comments to the Author” section, enter your conflict of interest statement in the “Confidential to Editor” section, and submit your "Accept" recommendation.

Reviewer #3: All comments have been addressed

2. Is the manuscript technically sound, and do the data support the conclusions?

Reviewer #3: Yes

3. Has the statistical analysis been performed appropriately and rigorously? 

Reviewer #3: N/A

4. Have the authors made all data underlying the findings in their manuscript fully available?

Reviewer #3: Yes

5. Is the manuscript presented in an intelligible fashion and written in standard English?

Reviewer #3: Yes

6. Review Comments to the Author

Reviewer #3: All previous comments have been sufficiently addressed. The authors have acknowledged the value an external dataset would add for testing to support the presented results. Final comments are:

1) please add numbers of patients used (not just samples) for experiments 2 and 3 in methods section lines 202 and 205, respectively.

2) where relevant in the text, figures and figure legends please use gene names rather than the alphabetised classes representing genes. It is very awkward for the reader to have to check each time which letter represents which gene and would improve readability.

7. PLOS authors have the option to publish the peer review history of their article (what does this mean?). If published, this will include your full peer review and any attached files.

Reviewer #3: No

---

## [Author Response · Author response to Decision Letter 4]

15 Apr 2026

We sincerely thank the reviewer and the editorial board for the careful evaluation of our manuscript and for the positive assessment. We are pleased that the reviewer finds the manuscript to be technically sound and that the previous comments have been adequately addressed. Below, we provide point-by-point responses to the remaining minor comments.

Comments to the Author

1. If the authors have adequately addressed your comments raised in a previous round of review and you feel that this manuscript is now acceptable for publication, you may indicate that here to bypass the “Comments to the Author” section, enter your conflict of interest statement in the “Confidential to Editor” section, and submit your "Accept" recommendation.

Reviewer #3: All comments have been addressed

Response:

We thank the reviewer for this positive assessment and for the constructive feedback throughout the review process.

2. Is the manuscript technically sound, and do the data support the conclusions?

Reviewer #3: Yes

Response:

We appreciate the reviewer’s positive evaluation and are grateful for the constructive input, which has helped strengthen the manuscript.

3. Has the statistical analysis been performed appropriately and rigorously?

Reviewer #3: N/A

4. Have the authors made all data underlying the findings in their manuscript fully available?

Reviewer #3: Yes

Response:

We thank the reviewer for highlighting the importance of data and code availability. In accordance with PLOS ONE data sharing policies, we have updated the manuscript to include access to the code:

Line 124: Code is available at: https://github.com/timshik/Deep-learning-based-identification-of-inherited-retinal-disease-using-wide-field-retina-imaging.

Line 215: The implemented method is available at link: https://github.com/timshik/Deep-learning-based-identification-of-inherited-retinal-disease-using-wide-field-retina-imaging.

5. Is the manuscript presented in an intelligible fashion and written in standard English?

Reviewer #3: Yes

Response:

We thank the reviewer for this positive assessment.

6. Review Comments to the Author

Reviewer #3: All previous comments have been sufficiently addressed. The authors have acknowledged the value an external dataset would add for testing to support the presented results. Final comments are:

1) please add numbers of patients used (not just samples) for experiments 2 and 3 in methods section lines 202 and 205, respectively.

Response:

We thank the reviewer for this helpful suggestion, which improves clarity. We have now explicitly included the number of patients alongside the number of images in the Methods section.

The text has been revised as follows:

Line 202 (Experiment 2):

Experiment 2: Five-class classification task including only IRD patients belonging to Group 1 (ABCA4, NR2E3, RDH12, TRPM1, USH2A). A total of 108 patients (298 images) were included. Each patient was assigned to one of the five gene classes (mean test set size ~60 studies per fold).

Line 205 (Experiment 3):

“Experiment 3: Five-class classification task including only IRD patients belonging to Group 2 (BEST1, CHM, CNGA3, CRB1, FAM161A). A total of 74 patients (196 images) were included. Each patient was assigned to one of the five gene classes (mean test set size ~39 studies per fold).”

2) where relevant in the text, figures and figure legends please use gene names rather than the alphabetised classes representing genes. It is very awkward for the reader to have to check each time which letter represents which gene and would improve readability.

Response:

We fully agree with the reviewer that this change improves readability and reduces unnecessary back-and-forth for the reader. We have revised the manuscript accordingly by replacing alphabetized class labels with the corresponding gene names throughout the text, figures, and figure legends wherever applicable

---

## [Editor Report · Decision Letter 4]

23 Apr 2026

Using deep learning to identify inherited retinal diseases based on wide-field retinal imaging data

PONE-D-24-51430R4

Dear Dr. Levy,

We’re pleased to inform you that your manuscript has been judged scientifically suitable for publication and will be formally accepted for publication once it meets all outstanding technical requirements.

Kind regards,

Ogugua Ndubuisi Okonkwo, M.D.

Academic Editor

PLOS One

---

## [Editor Report · Acceptance letter]

PONE-D-24-51430R4

PLOS One

Dear Dr. Levy,

I'm pleased to inform you that your manuscript has been deemed suitable for publication in PLOS One. Congratulations! Your manuscript is now being handed over to our production team.

Kind regards,

on behalf of

Prof Ogugua Ndubuisi Okonkwo

Academic Editor

PLOS One